# Emotional Health: Improving Emotional Intelligence Through Physical Education

**DOI:** 10.3390/healthcare13192540

**Published:** 2025-10-09

**Authors:** Karen Troncoso-Ulloa, Izaskun Luis-de-Cos, Saioa Urrutia-Gutierrez, Gurutze Luis-de-Cos, Silvia Arribas-Galarraga

**Affiliations:** 1Department of Education, University of Antofagasta, Antofagasta 1240000, Chile; karen.troncoso@uantof.cl; 2Department of Musical, Visual Arts and Physical Education, University of the Basque Country (UPV/EHU), 48940 Leioa, Spain; izaskun.luis@ehu.eus (I.L.-d.-C.); saioa.urrutia@ehu.eus (S.U.-G.); 3Department of Health Sciences, Public University of Navarre (UPNA), 31006 Pamplona, Spain; gurutze.luis@unavarra.es

**Keywords:** emotional intelligence, emotional physical education, mental health, students

## Abstract

**Background/Objectives**: Difficulties in emotional regulation are recognized as a risk factor for a variety of emotion-based psychopathologies, including anxiety and depression. In this context, movement and physical activity have been identified as a key element in preventing these health issues, particularly during the initial teacher training. This study aimed to analyze the impact of an emotional health physical education program on the Emotional Intelligence of university students studying physical education in Chile. **Methods**: A quasi-experimental design with experimental and control groups and repeated measures (pre test-post test) was employed. A total of 214 male and female students from two Chilean universities participated and completed the Spanish version of the Trait Meta-Mood Scale (TMMS-24). **Results**: Results confirmed the program’s effectiveness in fostering Emotional Intelligence, revealing statistically significant improvements (*p* < 0.05) in the dimensions of emotional attention and clarity. **Conclusions**: These findings suggest that emotional physical education programs can be effective in promoting emotional skills essential for the learning and mental well-being of university students who will later become primary and secondary school teachers.

## 1. Introduction

Emotional health has become a pressing concern in today’s society, affecting individuals, regardless of age or status. The educational field, given its universal nature, faces new challenges and acts as an ideal environment to promote the development of emotional competences involved in resolving conflicts and overcoming challenges.

During initial teacher training, the teaching–learning process must be adapted to diverse social and cultural contexts, and therefore students must be equipped with social and emotional skills that enhance their adaptability to new situations [1,2]. As Suira [3] points out, teachers must learn to manage stress effectively to ensure student’s lifelong learning and improve their quality of life. Therefore, emotional development plays an essential role not only for comprehensive training and personal growth (Barranco-Moreno et al. [4]), but also for preparing future educators to manage classroom dynamics and prevent mental health issues stemming from poor emotional management or low Emotional Intelligence [5].

Physical activity and movement have been identified as significant factors against mental health disorders, with research showing that physical activity directly influences mental well-being [6,7]. Engagement in physical activity has been associated with reduced emotional and behavioral problems [8].

Emotions serve as central component of information processing and decision-making, directly influencing personal success across all areas of life [9,10,11]. This highlights their fundamental importance in personal development and supports the integration of emotional training in initial teacher education programs. The ability to recognize, understand, and regulate emotions not only enhances psychological well-being, but may also contribute to overcoming difficulties in emotional regulation. Larinow et al. [12], identify emotional dysregulation as a risk factor for a wide range of emotion-based psychopathologies, including anxiety disorders and depression. In contrast, individuals with higher emotional intelligence tend to report lower levels of psychological distress, enhanced resilience, and better coping strategies when facing stressful situations [13]. Strengthening emotional intelligence therefore contributes to greater emotional stability, increased self-esteem, and more adaptive interpersonal functioning—key components of emotional health [14,15,16].

Emotional Intelligence is a fundamental concept in the field of psychology and personal development. Mayer and Salovey [17] defined Emotional Intelligence as “the ability to perceive, appraise, and express emotions accurately; the ability to understand emotions and emotional knowledge; and the ability to regulate emotions in ways that promote emotional and intellectual growth” (p. 10). Developing Emotional Intelligence is considered essential for personal growth and successful interpersonal relationships [18]. According to Bisquerra et al. [10] and Sánchez-Ortega and Chacón-Cuberos [19], emotional development must be continuous to be truly effective, fostering emotional expression, empathy, understanding, and reflective thinking. These competences should be developed in different scenarios that promote experimentation and analysis of emotions, thereby contributing to a healthy environment.

Several authors [20,21,22] claim that intervention programs using participatory and engaging methodologies may be the most effective for fostering and developing Emotional Intelligence in the educational environment, as they integrate both cognitive and experimental elements [23].

There are several studies in the scientific literature on the effectiveness of Emotional Intelligence education programs based on physical activity/physical education. The study carried out by Castillo Viera et al. [24], using physical education content, concluded that dramatization led to improvements in certain factors of Emotional Intelligence in female students, and especially in male students. Additionally, the study by Pozo-Rosado et al. [25], which analyzed the effects of a program based on Teaching Personal and Social Responsibility on empathy and perceived Emotional Intelligence in physical education in different socio-economic contexts, concluded that the program was especially effective in improving Emotional Intelligence among students from socioeconomically disadvantaged backgrounds.

While there is some literature on school-based Emotional Intelligence development programs (García-Merino and Lizandra [26]; Pellicer [23]), such programs remain scarce in the university context. In this context, there is a clear need to develop strategies that foster emotional health, responding to an idea that has been gaining interest: the possible relationship between emotions and learning.

This study is particularly relevant because, following evidence obtained in previous studies [23,26], emotional education is employed to achieve healthy outcomes in future teachers, both through its direct impact on them and through the multiplier effect that could be generated when those who now receive this training go on to pursue their professional careers, potentially benefiting around 30 children for every student who has participated in this program.

Among the few studies conducted in the university context, the study by Luis de Cos et al. [1] stands out, as they had proposed a didactic intervention based on physical education with the aim of developing socio-emotional competences of Primary Education students at the University of the Basque Country. The present study takes this intervention as a reference and adapts it for implementation with Latin American population.

Therefore, in order to address the challenges of quality education and fostering an emotionally healthy life and well-being, this study evaluates the impact of a healthy emotional physical education program on designed to promote emotional recognition and identification, emotional expression and regulation, and empathy. The aim is to determine how these practices contribute to emotional health through the emotional and personal growth of the participants.

## 2. Materials and Methods

### 2.1. Participants

The study sample consisted of 214 male and female students from two universities: Universidad de Antofagasta (experimental group) and Universidad Autónoma de Chile (control group). Regarding gender, 68.7% were men and 31.3% were women, aged between 18 and 25 years.

A convenience sampling method was used. To ensure comparability between the participating universities, it was established that both universities offered a degree in physical education with a similar curricular structure, both in content and duration (10 semesters). Additionally, both control and experimental groups were required to be enrolled in courses with comparable curricula and located in the same academic semester.

The designation of control and experimental groups, was determine by the availability of resources and time commitment of the teaching staff, assigning the University of Antofagasta as the experimental group and the Universidad Autónoma de Chile as the control group.

Exclusion criteria included non-participation in any of the practical sessions of the courses in which the program was developed and failure to complete the pre- or post-questionnaires.

### 2.2. Design and Instrument

This research was preceded by a pilot study conducted with university students at the UPV-EHU during the 2022–2023 academic year, after which aspects such as clarity of instructions and protocols for the implementation of some of the practical sessions were refined. Likewise, some of the variables (learning style and mental toughness) that were not relevant to the study were eliminated. Following these adjustments, the project was redesigned, leading to the current study.

A quasi-experimental design was employed, incorporating experimental and control groups and repeated measures (pre-test–post-test).

The intervention program was the independent variable, while Emotional Intelligence was the dependent variable.

To measure Emotional Intelligence, the Spanish version of the Trait Meta-Mood Scale (TMMS-24) by Salovey, Mayer, Goldman, Turvey, and Palfai [27], validated in the Spanish population [28], was employed to study the dependent variable (Emotional Intelligence). The TMMS-24 scale is a self-report scale composed of 24 items that measure beliefs about one’s own Emotional Intelligence. The scale consists of three subscales that assess the following: (a) pay attention to and value their feelings (Attention: e.g., “I think about my mood constantly”), (b) feel clear rather than confused about their feelings (Clarity: e.g., “I almost always know exactly how I feel”), and (c) use positive thinking to repair negative moods (Repair: e.g., “Although I am sometimes sad, I have a mostly optimistic outlook”). The 24 items are answered on a five-point Likert scale, ranging from 1 “Strongly disagree” to 5 “Strongly agree”.

### 2.3. Procedure

The study adhered to ethical principles for research involving humans. It was approved by the Committee on Ethics in Scientific Research of the University of Antofagasta, Chile (CEIC-UA) under folio number 435/2023, as outlined in the Nuremberg Code, the Declaration of Helsinki, CIOMS, and according to the guidelines of Ezekiel J. Emanuel.

To collect the data, both universities were contacted, and once the dates for survey administration (Universidad Autónoma de Chile) and intervention implementation (Universidad de Antofagasta) were confirmed, the study commenced.

Before the intervention, the participants were informed about the characteristics of the project. Researchers provided a brief introduction outlining the objective and addressing any student inquiries. Students who agreed to participate were asked to complete an informed consent form. Subsequently, the pretest was administered to both experimental and control groups, using the TMMS-24.

Upon collection of consent forms, the intervention program was implemented in the experimental group. This involved five specific sessions focused on emotions through body expression, and a transversal work throughout a 14-week course). Meanwhile, the control group received the standard program. After the intervention period, the same instrument used in the pre-test was administered to both experimental and control groups.

### 2.4. Intervention

The emotional health physical education program (Table 1) was based on Bisquerra’s Pentagonal Model of Emotional Competences and was carried out over a 14-week period, with a total of 10 direct intervention sessions. These included four full 90 min sessions and six shorter sessions lasting between 20 and 30 min. The methodology was experiential, practical, group-oriented, dynamic, and participatory, emphasizing respect, freedom, and autonomy for the participants [29].

This program for emotional health considers physical-educational activity as a central medium for emotional development, allowing students to experience emotional exchanges and to perceive, understand, and express emotions [30]. It facilitates the progressive development of emotional competences, beginning with emotional awareness, regulation, and autonomy, and advancing to more complex skills such as social competence and daily life abilities.

### 2.5. Data Analysis

The SPSS 27th version was used to do the analysis of the data. Initially, data normality was assessed using the Kolmogorov–Smirnov test, which confirmed that followed a normal distribution. Homogeneity of variances was verified using Levene’s test. In all cases the *p*-value exceeded 0.05, confirming assumption of equal variances. Therefore, parametric statistical methods were applied.

Paired-samples *t*-tests were conducted to examine differences between the scores obtained by the students in each group in the pre-test and post-test. Effect sizes were calculated using Cohen’s d to determine the magnitude of these differences [31].

Additionally, independent-samples *t*-tests were conducted to compare the scores and gains in three dimensions of EI between control and experimental groups. Cohen’s d was also used to determine the magnitude of the differences.

According to Cohen, effect size is interpreted as follows: small (0.2–0.49), medium (0.5–0.79), and large (≥0.8).

## 3. Results

Table 2, Table 3 and Table 4 present the pre- and post-intervention scores, as well as the progress in Emotional Intelligence among students in both experimental and control groups. For each score, the mean (M) and standard deviation (SD) are reported.

Table 2 shows the results for Attention subscale of the Emotional Intelligence, calculated as the difference between post-test pre-test scores. A paired-samples *t*-test revealed statistically significant differences in the experimental group between pre-test and post-test scores, with a small effect size (Cohen’s d = 0.30). These findings suggest that students who participated in the program demonstrated a significant improvement in their competence to attend to their emotions. In contrast, no significant differences were found in the control group.

Continuing with Table 2, the scores of the two study groups were compared using *t*-test for independent samples. A statistical difference was found in the post-test scores, which were higher in the experimental group, with a small effect size (Cohen’s d = 0.3).

Table 3 presents the results for the Clarity subscale of Emotional Intelligence. A paired-samples *t*-test indicated statistically significant differences in the experimental group between pre-test and post-test, with a small effect size (Cohen’s d = 0.3).

Regarding the comparison between pre-test and post-test in both experimental and control group, a statistically significant difference was also observed, with the control group displaying lower scores. The effect size was small (Cohen’s d = 0.24).

Finally, Table 4 displays the results for Repair subscale of Emotional Intelligence. No significant differences were found between the pre-test and post-test scores in either the experimental or the control group. Similarly, no significant differences were observed between the experimental and control groups at either the pre-test or the post-test.

## 4. Discussion

The results presented address the main objective of the study: to evaluate the effectiveness of a physical education-based program in developing Emotional Intelligence among university students. Findings confirm the program’s effectiveness, specifically in the dimensions of emotional attention and clarity.

The data suggest that the intervention had a positive and statistically significant effect on students’ emotional attention. Participants in the experimental group presented an adequate level of attention to their emotions and marked improvement compared to the control group. This improvement in their emotional attention enables students to better perceive and recognize their own emotions, fostering greater self-awareness and facilitating the identification and acceptance of emotional states without ignoring them or becoming overwhelmed [15,32].

Similarly, enhanced emotional clarity indicates that the program enabled students in the experimental group to better understand and differentiate their emotions. Greater emotional clarity enables them to more accurately interpret what they are feeling and to distinguish between different emotional states [33,34].

The development of these dimensions not only fosters awareness of one’s own emotions but also improves the capacity to recognize emotions in others, thereby facilitating the perception of the emotional climate within social or educational context. This ability is essential for developing empathy in interpersonal relationships, a primary skill for their future professional practice. In this way, this emotional awareness enables them to better understand the interaction between emotions and behavior [35], and to develop stronger classroom dynamics that can help prevent emotional stressful situations [3].

In contrast, the absence of significant improvement in emotional repair dimension may be attributed to the limited duration of the program, which might not have been enough to cultivate more complex emotional regulation skills. Supporting this interpretation, Sigüenza-Marín et al. [36], suggest that longer programs are more likely to produce effects on emotional competencies. Similarly, Puertas-Molero et al. [32] report that programs lasting between four and eleven months tend to be more effective.

Thus, the present study aligns with previous research indicating that emotional learning through physical-sports practice can lead to meaningful development of certain dimensions of Emotional Intelligence [37,38]. Development of these dimensions enables students to gain deeper self-awareness, identify and understand their emotions, and finally engage in more empathetic and effective interpersonal communication. These highlight the importance of implementing structured and emotionally focused interventions, within physical education context.

From a broader perspective, the development of Emotional Intelligence, due to its significant impact on emotional health and well-being, contributes positively to mental health. Prior studies Brackett et al. [39]; Fernández Berrocal et al. [40]; Martínez González et al. [41]; Suira [3] have shown that the development of emotional competencies is associated with reduced levels of stress and anxiety, improved interpersonal relationships, and enhanced ability to manage emotions effectively and respond to stressful situations with greater resilience.

It can be concluded that one of the key factors contributing to the success of the program may be its methodology approach. The intervention followed the pedagogical guidelines proposed by Bisquerra [20] and Pellicer [23], emphasizing experiential, student-centered learning. The program based on physical education includes meaningful strategies such as mindfulness, role-playing, relaxation, emotional expression through the body, and cooperative aerobics. These activities are combined with individual and group support, reflection, and introspection through an emotional diary.

The effectiveness of this program are consistent with trends observed in previous research, such as Castillo Viera et al. [24], Luis-de-Cos et al. [1], and Postigo-Zegarra et al. [42], who demonstrated the positive impact of educational programs on emotional development. In line with Bisquerra [21] and Castillo et al. [22], interventions based on emotional physical education—when rooted in practical, reflective, and student-centered approaches—are particularly effective for fostering Emotional Intelligence.

Finally, this study is not without limitations. The use of a quasi-experimental approach involving students from different universities may introduce biases or uncontrolled factors beyond the scope of the study. Additionally, because of the relatively short duration of the intervention, it was not possible to fully achieve isolation of all external variables, which may also contribute to potential biases or unwanted effects. To address these limitations, future research should consider implementing the intervention in a common educational context, controlling external variables as much as possible and with longer duration.

Despite these limitations, this empirical research, together with previous studies, reinforces the relevance of the main findings: an emotional health physical education program represents a valuable strategy to improve Emotional Intelligence in future teachers.

## 5. Conclusions

Therefore, it can be concluded that the intervention based on healthy emotional physical education had a positive and significant impact on the improvement of the participants’ attention and clarity, thus improving their Emotional Intelligence. It also reveals the importance of carrying out interventions in the classroom that promote the development of Emotional Intelligence and its competencies, favoring the development of tools for the well-being and health of future teachers both in their training phase and in their future performance.

This study is particularly relevant because, at theoretical level, it reinforces the precepts addressed by authors such as Bisquerra [21], Garcia-Merino and Lizandra [26], and Pellicer [23]. Through the evidence gathered in this research, not only are the benefits of implementing strategies to improve the emotional health of future teachers demonstrated, but also the multiplier effect of these benefits. In other words, the positive impact experienced by participating students may be transferred to their future students. This emphasizes the broader educational value of designing emotional education programs based on physical education into teacher preparation programs.

## Figures and Tables

**Table 1 healthcare-13-02540-t001:** Summary of emotional health physical education program. Modified from Luis de Cos et al. [1].

Session	Competition	Activity to Be Carried Out	Components	Target
**Session 1**	Emotional Awareness	Introduction to emotional physical education	Emotional Intelligence and its models, education, and emotional competences.	**Recognize and differentiate** the concepts of intelligence, education, and emotional competences, relating them to their teacher training and daily life.
**Session 2**	Emotional Awareness	Body expression activities	Recognition and understanding of one’s own emotions and those of others.	Recognize and understand one’s own and others’ emotions through the development **of body language.**
**Session 3**	Emotional Regulation	Competitive vs. cooperative games	Emotional management, coping skills and frustration tolerance.	Development of emotional regulation **through competitive recreational activities**.
**Session 4**	Social Competence	Cooperative workshop; feel, act and share	Responsive communication/pro-social behavior and cooperation.	Develop social competence **through cooperation and team-work to achieve common goals**.
**Session 5**	Emotional Awareness	Body expression activities	Recognition and understanding of one’s own emotions and those of others.	Recognize and respect the emotions of others and oneself, **through the expression of an emotional experience and body language.**
**Session 6**	Emotional Regulation	Relaxation, breathing techniques	Self-generation of positive emotions	Balance physiological and environmental variables to self-generate positive emotions, **through mindfulness.**
**Session 7**	Emotional Autonomy	Role-playing activities	Self-esteem/positive attitude: Self-confidence	Valuing one’s own personal qualities, expressing positive qualities of others **and developing a positive attitude toward life.**
**Session 8**	Social Competence	Role-playing activities	Pro-social behavior/expressive and receptive communication	Enhance **expressive and receptive communication** to develop **prosocial and cooperative behavior**
**Session 9**	Social Competence/Life Skills and Assets	Gamification Proposal	Types of communication: assertive, passive, and aggressive	Experiencing the different forms of communication (assertive, passive, and aggressive), promoting the use of **assertive communication.**
**Session 10**	Life skills and well-being	Role-playing activities	Decision-making	Developing **negotiation skills** to achieve joint challenges. Emotion regulation through role-playing.

**Table 2 healthcare-13-02540-t002:** Pre-test and post-test results for attention (Emotional Intelligence).

Attention	Experimental	Control	Independent Samples, *t*-Test *p*-Value	D for Cohen
	(N = 108)	(N = 106)		
Pre-test [M (SD)].	28.63 (6.07)	27.73 (6.13)	0.14	0.14
Post-test [M (SD)].	30.31 (5.67)	28.47 (5.56)	0.009 *	0.3
Paired samples, *t*-test *p*-value	0.001 *	0.62	-	-
D de cohen, size effect	0.3	0.15	-	-

* *p* < 0.05.

**Table 3 healthcare-13-02540-t003:** Pre-test and post-test results for clarity (Emotional Intelligence).

Clarity	Experimental	Control	Independent Samples, *t*-Test *p*-Value	D For Cohen
	(N = 108)	(N = 106)		
Pre-test [M (SD)].	27.76 (5.98)	29.97 (5.50)	0.416	0.02
Post-test [M (SD)].	29.66 (6.09)	28.16 (6.24)	0.038 *	0.24
Paired samples, *t*-test *p*-value	0.001 *	0.335	-	-
D de cohen, size effect	0.3	0.042	-	-

* *p* < 0.05.

**Table 4 healthcare-13-02540-t004:** Pre-test and post-test repair results (Emotional Intelligence).

Repair	Experimental	Control	Independent Samples, *t*-Test *p*-Value	D for Cohen
	(N = 108)	(N = 106)		
Pre-test [M (SD)].	29.44 (5.39)	29.97 (5.50)	0.24	0.09
Post-test [M (SD)].	29.46 (6.46)	29.83 (5.78)	0.327	0.06
Paired samples, *t*-test *p*-value	0.487	0.408	-	-
D de cohen, size effect	0.003	0.023	-	-

## Data Availability

Data is contained within the article.

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
