# Peer review of "Emotional Health: Improving Emotional Intelligence Through Physical Education"

_healthcare, 2025, doi:10.3390/healthcare13192540_

Round 1

Reviewer 1 Report

Comments and Suggestions for Authors

This study performed a quasi-experimental study to examine possible effects of a healthy emotional physical education program on the development of emotional intelligence in Chilean university students. Results showed significant positive effect of such intervention in developing emotional intelligence with improvements of emotional attention and clarity.

Overall, the present manuscript focuses on an important issue the research question is sound and that might be of interest of Healthcare readers. Despite these positive aspects, there are a few issues that need to be clarified, reviewed and improved in the manuscript. These issues are listed below.

Major issues:

The presentation of the goal could be improved. The aim presented in the abstract is well defined, but this is not the case at the end of the introduction (lines 93-99). In this paragraph, the organization and presentation is confusing and gives the impression that the study has two purposes. Please revise it and improve for clarity.

It was mentioned that the present manuscript used as reference the intervention of the Luis de Cos et al. because it “stands out”. Why does it stand out? Please further develop the reasoning here for clarity for the readers.

Procedures:

  • Please provide further information about the two sites in which the participants were from. Were these two Universities comparable? How the experimental and control were defined? Please clarify this important aspect.
  • Statistical analysis: why t-tests were employed instead of mixed-model Analyses of Variance?
  • Finally, t-test does not require normality of the data: it uses its own distribution. On the other hand, when employing independent samples t-tests, it is necessary to make sure that homogeneity of variance is fulfilled: was it verified?

Minor issues:

  • Please verify citation format.

Please revise for English formatting and editing.

Comments on the Quality of English Language

English needs to be revised for editing.

Reviewer 2 Report

Comments and Suggestions for Authors

Dear authors,

The article is innovative and addresses a current and challenging issue.
The introduction is very general, and the importance of the constructs analyzed and their application in a university physical education program is not evident.
I suggest including a literature review section. This will allow for a specific presentation of the conceptual and operational definitions of the variables, dimensions of analysis, and their application.
Because this is a quasi-experimental design, I suggest also including the participant exclusion criteria in the materials and methods section.
The conclusion lacks robustness. There is no special emphasis on the importance of the study. I suggest specifically including the theoretical and practical implications of the study.

Kind regards,

Reviewer 3 Report

Comments and Suggestions for Authors

Dear author/s

I congratulate the authors for their efforts. I made some suggestions for the development of the study.

- Emotional health term is quite ambitious and exaggerated for academic title especially educational study. Emotional regulation, emotional skills etc. should have been preferred.

- The paragraph of importance of the study offers a decent rationale. What does the study suggest to fill the gap? You should have given more space to this.

- You should have highlighted why you work with physical education students.

- Who decided on the appropriateness of your intervention program? No pilot study was conducted. Skipping a pilot study and proceeding directly to the experimental phase involves certain methodological risks. Without preliminary testing, problems such as unclear instructions, timing issues, or unexpected participant reactions may go unnoticed. This can negatively affect data quality and internal validity. A pilot study helps identify and resolve such issues early, ensuring the main study runs smoothly and reliably.

- How did you separate the experimental and control groups? What was your criterion? Should you elaborate further?

- The connection between emotional health and emotional intelligence is not fully explained in either the introduction or the discussion section.

- It would not be wrong to say that the implications in the study is insufficient and weak.

Comments on the Quality of English Language

You should proofreading.

Reviewer 4 Report

Comments and Suggestions for Authors

Dear Authors,

Thank you for the opportunity to review your manuscript. Below are my detailed comments and recommendations, organized by section:

Abstract
The abstract follows a standard structure and provides a concise overview of the study. However, I recommend including specific statistical data to support the phrase "...showing significant improvements..." in order to substantiate the claim with evidence. Additionally, in the "Key words" section, the presence of the number "1" is unclear and appears to be a formatting error. I also suggest listing the keywords in alphabetical order for clarity and consistency.

Introduction
The introduction is well written and supported by 24 citations. However, the oldest citation dates back to 1997. I recommend replacing this with a more recent source, if available, to maintain the timeliness of your references. Regarding citation formatting, while most references are correctly placed in square brackets, there is one error on line 50, which should be formatted as "9–11" (e.g., consistent with formatting on line 58: "13–14" and line 70: "18–20"). The objective of the study is clearly and succinctly stated.

Materials and Methods
This section is written adequately overall. The "Procedure and Intervention" subsection is well described and requires no changes. However, I strongly recommend expanding the sections on "Participants," "Design and Instrument," and "Data Analysis" to provide a more thorough description of your methodology. Additionally, numbering the subsections (e.g., 2.1 Participants, 2.2 Design and Instrument, etc.) would improve readability and organization.

Results
The results section is concise, perhaps too brief. The tables are clear and well-structured, but there is an issue with the number formatting. Please replace commas with periods in decimal values (e.g., change 0,14 to 0.14 and 0,009 to 0.009) to conform with standard scientific conventions.

Discussion
This section is comprehensive and adequately supported by citations. The discussion thoughtfully interprets the results and appropriately addresses both limitations and future research directions. It demonstrates a solid understanding of the implications of the findings.

Conclusions
The conclusion is brief and to the point. However, I recommend slightly expanding this section to restate the most important findings. Including a key statistical result would reinforce the significance of the study’s outcomes.

Recommendation
Reconsider after major revisions. Substantial improvements are needed, particularly in the Methods and Results sections, as well as in the abstract and conclusion.

Kind regards,
Reviewer

Round 2

Reviewer 4 Report

Comments and Suggestions for Authors

No comments.